# Characterization of Skin Interfollicular Stem Cells and Early Transit Amplifying Cells during the Transition from Infants to Young Children

**DOI:** 10.3390/ijms25115635

**Published:** 2024-05-22

**Authors:** Marika Quadri, Caroline Baudouin, Roberta Lotti, Elisabetta Palazzo, Letizia Campanini, François-Xavier Bernard, Gaëlle Bellemere, Carlo Pincelli, Alessandra Marconi

**Affiliations:** 1DermoLab, Department of Surgical, Medical, Dental and Morphological Sciences, University of Modena and Reggio Emilia, 41124 Modena, Italy; marika.quadri@unimore.it (M.Q.); elisabetta.palazzo@unimore.it (E.P.); carlo.pincelli@unimore.it (C.P.); alessandra.marconi@unimore.it (A.M.); 2Expanscience Laboratoires, 28230 Eprernon, France; cbaudouin@expanscience.com (C.B.); gbellemere@expanscience.com (G.B.); 3Bioalternatives, 86160 Gencay, France; fxb@bioalternatives.com

**Keywords:** skin barrier, keratinocyte stem cells (KSC), early transit amplifying (ETA) cells, neonatal skin, pediatric skin

## Abstract

In the interfollicular epidermis, keratinocyte stem cells (KSC) generate a short-lived population of transit amplifying (TA) cells that undergo terminal differentiation after several cell divisions. Recently, we isolated and characterized a highly proliferative keratinocyte cell population, named “early” TA (ETA) cell, representing the first KSC progenitor with exclusive features. This work aims to evaluate epidermis, with a focus on KSC and ETA cells, during transition from infancy to childhood. Reconstructed human epidermis (RHE) generated from infant keratinocytes is more damaged by UV irradiation, as compared to RHE from young children. Moreover, the expression of several differentiation and barrier genes increases with age, while the expression of genes related to stemness is reduced from infancy to childhood. The proliferation rate of KSC and ETA cells is higher in cells derived from infants’ skin samples than of those derived from young children, as well as the capacity of forming colonies is more pronounced in KSC derived from infants than from young children’s skin samples. Finally, infants-KSC show the greatest regenerative capacity in skin equivalents, while young children ETA cells express higher levels of differentiation markers, as compared to infants-ETA. KSC and ETA cells undergo substantial changes during transition from infancy to childhood. The study presents a novel insight into pediatric skin, and sheds light on the correlation between age and structural maturation of the skin.

## 1. Introduction

The skin is the first line of defense protecting the body from environmental factors [1]. Functional and structural skin adaptation is an ongoing dynamic process in the first year of postnatal life following adjustment to an extrauterine dry environment. Skin acidification, water handling properties and permeability are interconnected postnatal modifications that take place in the outer stratum corneum skin layer [2,3,4,5,6]. In vivo data revealed that the skin surface evolves with age from the postnatal period into early childhood, as biochemical maturation signals are produced, and stratum corneum water handling capacity is affected [4,5]. Markers of skin maturation from birth to adulthood have been described with electron microscopy techniques, reporting a clear correlation between age and structural maturation of the stratum corneum. The report confirmed the relative immaturity of the epidermal barrier up to two years after birth under basal conditions [6].

Keratinocyte stem cells (KSC) reside in the basal layer (niche), govern the renewal of epidermis and generate transit amplifying (TA) cells that terminally differentiate after a number of cell divisions [7,8]. TA cells are responsible for most of the proliferative activities and are capable of extensive expansion of the cell population [9]. A small change in the number of cell divisions that a TA cell can undergo could have dramatic effects on the final cell population [10,11]. We have recently isolated a highly proliferative epidermal basal cell, named “early” TA (ETA) cell, based on its capacity to adhere to type IV collagen, that represents the first KSC progenitor with exclusive features [11].

To improve our knowledge of skin maturation and epidermal keratinocyte physiology during infancy and early childhood, we investigated skin barrier functionality against UV, epidermal differentiation, stemness and regeneration in keratinocyte subpopulations (KSC and ETA) from infants and young children using 2D and 3D culture models.

## 2. Results

### 2.1. Epidermis in Infants and Young Children

Because in infancy epidermis appears to be immature and the barrier not fully protective, we wanted to compare reconstructed human epidermis (RHE) generated from infants and young children after UV irradiation. RHEs presented a normal structure and displayed the expected morphology on day 12, regardless of donor age (0–2 years or 3–6 years). However, the infant RHE was more strongly affected by UV irradiation than the RHE from young children. UV irradiation caused a clear separation of stratum corneum in the RHE from infants which was not observed in the RHE from young children (Figure 1a). Sunburn cells were observed in both irradiated RHEs, with a higher proportion in the infant epidermis, as compared to young children epidermis (57.4 ± 1.14/mm^2^ vs. 52.9 ± 1.58/mm^2^) (Figure 1b).

To further evaluate differences between infants and young children epidermis, we analyzed the expression of genes involved in epidermal development/differentiation and skin barrier function. Claudin 1, BARX2 and Sciellin were scarcely expressed in infants, while their expression significantly increased in young children (Figure 1c). On the other hand, the relative expression of interfollicular epidermal stem cell genes (Keratin 19, α_6_-integrin, β_1_-integrin, β_4_-integrin, NOTCH 1 and Nidogen) was highest in infants, and then decreased with age, as shown in the young children group (Figure 1d).

### 2.2. Stemness and Differentiation of KSC and ETA Cells in Infants and Young Children

Given that little is known about the features of the keratinocyte subpopulations in infancy, we evaluated the protein expression of different markers in KSC and ETA cells in infants up to 2 years of age, as compared to 3–6-years old young children. The expression of markers associated to stemness/proliferation tended to decrease with the increasing age of the donors. In particular, the number of β_1_-integrin, α_6_-integrin, ΔNp63, Notch1 and Keratin 15 (K15) positive cells was significantly lower in freshly isolated KSC derived from young children’s skin than of those derived from infants (Figure 1e and Appendix A). By western blotting, we observed a decrease in the expression of β_1_-integrin, ΔNp63 and Survivin in KSC derived from skin samples of 3–6 years as compared to those derived from 0–2 years skin (Figure 1g). Conversely, only β_1_-integrin and K15 positive cells were significantly modulated in ETA cells, with a significant decrease from infant to young children (Figure 1f and Appendix A). On the other hand, the number of Keratin 1 (K1) positive cells was lower in ETA derived from infants’ skin samples than in ETA from young children, while no expression of this early differentiation marker was noted in KSC cells (Figure 1e,f and Appendix A). K10 also increased significantly in the ETA cells of young children group, by western blotting analysis. As expected, the late differentiation markers Involucrin and Claudin 1 were expressed only in ETA cells. In detail, the number of Claudin 1-positive cells and Involucrin increased with age in ETA cells, although the increase of Involucrin is not statistically significant (Figure 1e,f and Appendix A). These data were confirmed by western blotting analysis, where increases of Claudin 1 were statistically significant (Figure 1h).

### 2.3. Proliferation and Clonal Capacity of KSC and ETA Cells in Infants and Young Children

While KSC proliferated to a highest rate in culture at 120 h, this function was significantly reduced in young children, as compared to infants (Figure 2a). Furthermore, colony forming efficiency (CFE) in KSC of young children was significantly reduced as compared to CFE in infants, with particular regard to larger colonies (>5 mm^2^) (Figure 2b). Likewise, ETA cells proliferated at a lower rate in young children than in infants at late time points (Figure 2c). In particular, the degree of cell proliferation in ETA from young children group was significantly less than in ETA from infants 5 days after cell plating. On the other hand, no difference between age groups was detected in CFE of ETA cells (Figure 2d).

### 2.4. Tissue Regeneration Ability of KSC and ETA Cells in Infants and Young Children

To test the tissue regeneration ability of KSC and ETA cells derived from infants and young children skin samples was used the full-thickness skin model, which efficiently recapitulates the epidermal and dermal compartment as in native skin tissues. Keratinocyte subpopulations such as KSC and ETA cells can generate 3D skin reconstructs, though to a different extent [11,12]. After 12 days, skin equivalents originated from infant KSC were compared to skin equivalents originated from young children KSC. Both skin equivalents showed a morphologically normal epidermis with a well-organized and polarized basal layer and differentiated suprabasal layers (Figure 2e). However, epidermal thickness significantly decreased during aging and Ki67 was significantly more expressed in skin equivalents from infant-KSC than from young children’s cells (Figure 2f). However, the age of KSC apparently does not affect differentiation/proliferation marker (NOTCH1, Involucrin and K15) expression in a significant manner, except for K1 and E-FABP, where expression significantly increased with age in KSC derived skin reconstructs (Figure 2g,h).

Similarly, skin equivalents both from infants and young children ETA cells showed a morphologically normal epidermis with a well-organized and polarized basal layer and differentiated suprabasal layers (Figure 2e). No significant difference in epidermal thickness and Ki67 expression was observed in skin equivalents from infants’ and young children’s cells (Figure 2e,f). On the other hand, differentiation tends to increase with age in skin equivalents generated from ETA cells. In particular, Involucrin, K1 and E-FABP expression was significantly more pronounced in young children than in infants. As expected, Notch-1 and K15 expression was lower in ETA cells than in KSC generated skin equivalents, but with no difference between the two age groups (Figure 2g,h).

## 3. Discussion

During the first years of life, infant skin undergoes a maturation process of its barrier function, which involves the careful interplay of several processes [13]. Our data support the notion that construction of an appropriate and functional skin barrier requires a balance between keratinocyte cellular processes of proliferation, differentiation and apoptosis when the tissue is stressed [8,14,15]. The relative expression of genes involved in epidermal differentiation and skin barrier function (Claudin-1, Involucrin, Keratin-1, BARX2 and Sciellin) increases from infancy to young childhood, indicating that the skin barrier probably develops and organizes in the first phase of age. The way stratum corneum stores and transports water and the maturation of normal stratum corneum desquamation and metabolic activities take years to fully reach adult levels [16]. Defects in keratinocyte differentiation and skin barrier are important features of inflammatory skin diseases such as atopic dermatitis, which usually starts during early infancy [17]. Our results correlate with previous in vivo results demonstrating that the skin barrier matures during early childhood [3,4,5].

With advancing age, epidermal stem cells dedicated to generating new differentiated cells lose their regenerative capacity, although their abundance is unaffected by skin aging [18]. Given the role of interfollicular stem cells in epidermal regeneration and in barrier function [19], it is not surprising that the stemness/proliferation markers (a_6_-integrin, ΔNP63, NOTCH1) are strongly expressed after birth (0–2 years of age). The gene expression of these markers decreases from infants to young children, whereas expression of epidermal differentiation and skin barrier function-associated genes increases.

After birth, the epidermis has to adapt to a new gaseous and dry environment, and a fine-tuning of its protective function is required. During this period of adaptation, stem cell resources are very likely mobilized for skin barrier construction. Skin reconstructs contribute to define the development of epidermis starting from keratinocytes of different age and with different differentiative state [20]. Here, our skin equivalent, performed starting from keratinocyte progenitors (KSC and ETA) derived from infants and young children skin samples, allows us to evaluate the development and formation of skin barrier in correlation to skin homeostasis. Therefore, we describe a period of relative epidermal immaturity during early childhood, and we observed an increase of differentiation (K1 and Involucrin) and skin barrier (E-FABP) associated markers from infants to young children.

The deleterious effects of UVs were more pronounced on the infant epidermis, in line with the greater sensitivity of infants’ and young children’s skin to sunlight, as seen by a stronger inhibition of barrier function and stem cell-related gene expression. In particular, among markers for KSC, Survivin has been shown to protect keratinocytes from UV-induced apoptosis [21]. Others have also reported an impaired capacity for protection and damage repair. Indeed, protection against the sun is central to avoiding skin cancer development, particularly in young children, since sunburns in childhood have been described as an important risk factor in the development of melanoma [22].

These results as a whole confirm the relative immaturity of the skin in infancy compared with young children and demonstrate that the epidermis becomes increasingly organized and competent during the first years of life. We are aware that the sample sizes for groups used in each assay could be a limit to highlight slight differences among ages, however, we strongly believe that our study demonstrates the relative immaturity of the skin in infants compared with young children, showing how the epidermis becomes increasingly organized and competent during the first years of life. In summary, we show that KSC and their progeny, critical for epidermal renewal, tend to decrease their potential with age. Infants’ and young children’s skin presents specific properties, in that relative expression of markers of epidermal differentiation and skin barrier increases and organizes with age, while the expression of stemness markers decrease. In RHEs, the differences observed in keratinocytes isolated from infants and young children are further exacerbated by external UV aggression, proving that skin is particularly vulnerable during early infancy. KSC and ETA cells, responsible for epidermal homeostasis, undergo substantial changes during transition from infant to young children in term of proliferation, differentiation/stemness and regenerative capacity. This study presents a novel insight into pediatric skin and reveals specific needs.

## 4. Materials and Methods

### 4.1. Human Keratinocytes Isolation

Normal human epidermal keratinocytes (NHEK) were isolated from surgical samples of infant (from 1 month to 2 years—group 0–2 years) and young children (from 3 to 6 years—group 3–6 years) healthy skin. Tissues were obtained from Tissue Biobank, CTIBiotech, France and from Policlinc of Modena, Italy. Informed consent was obtained from all subjects involved in the study. The study was conducted in accordance with the Declaration of Helsinki, and the protocol was approved by the Ethics Committee under registration at CTIBiotech, Meyzieu-Lyon, France (14 October 2013) and by Modena Medical Ethical Committee, Italy (Prot. 184/10). All infants and young children enrolled in the study are reported in Appendix A, indicating donors age and experiments performed on each skin sample.

### 4.2. KSC, ETA Cell Cultures and Skin Equivalents

Fresh keratinocytes were divided in KSC and ETA cells based on their ability to adhere to type IV collagen (100 μg/mL; Sigma, St. Louis, MO, USA), respectively for 5 and 15 min. In detail, keratinocytes were first allowed to adhere to type IV collagen for 5 min (KSC), and the non-adherent cells were then transferred to fresh collagen-coated dishes and allowed to attach for 15 min (ETA cells), as previously reported [11]. The two keratinocyte populations were cultured in serum-free medium (KGM-Gold Growth Medium, Lonza, Basel, Switzerland).

Skin equivalents were obtained by seeding KSC or ETA cells on dermal equivalents generated by fibroblasts-induced type I collagen contraction, as previously described [11].

### 4.3. Reconstruct Human Epidermis (RHE) and UV Irradiation

RHEs derived from total keratinocytes were cultured on an inert polycarbonate filter in Epilife^®^ culture medium (Cascade Biologics, Portland, OR, USA) with 50 µg/mL vitamin C [23]. RHEs were sham or irradiated with 500 mJ/cm^2^ UVB and 7255 mJ/cm^2^ UVA and again placed in fresh medium. The RHEs were then cultured 24 h and processed for H&E staining.

Sunburn cells were identified by their characteristic morphology that includes dark pycnotic and condensed basophilic nuclei, eosinophilic cytoplasm and intercellular spaces [24]. Sunburn cells were counted and compared with total surface of reconstructed epidermis under a Nikon E400 microscope (Nikon Europe B.V., Amstelveen, The Netherlands).

### 4.4. RT-PCR

CqRT-PCR was carried out on RNA of keratinocyte cultures from different donor age samples. After 48 h of incubation, the monolayer keratinocyte cultures were placed into TriPure Isolation Reagent^®^ and frozen at −80 °C. The quantity and quality of total extracted RNA were evaluated with a Bioanalyzer (Agilent Technologies, Santa Clara, CA, USA). Analysis of differential gene expression was evaluated by qRT-PCR using Glyceraldehyde-3-phosphate dehydrogenase (GAPDH) as a reference housekeeping gene. cDNAs were synthesized by reverse transcription in the presence of oligo dT and Superscript II enzyme (Gibco, Thermo Fisher Scientific, Waltham, MA, USA). The cDNA obtained was quantified by spectrophotometry and adjusted to 10 ng/µL. Quantitative PCR was conducted with the “light cycler” system (Roche Molecular System Inc., Pleasanton, CA, USA), as recommended by the manufacturer. Relative expression for each marker of interest was expressed in arbitrary units as follows: (1/2n) × 106, where n is the number of cycles.

### 4.5. Immunofluorescence of Isolated Cells

Freshly isolated KSC and ETA from infants and young children were fixed with 4% neutral buffered formalin, spotted onto glass slide and permeabilized with Triton-X100 0.1%. Cells were than incubated with primary antibodies: β_1_-integrin (1:50 dilution), α_6_-integrin (1:50 dilution), claudin 1 (1:100 dilution) (Abcam, Cambridge, UK); ΔNp63 (1:100 dilution), K1 (1:500 dilution) (BioLegend, San Diego, CA, USA); survivin (1:100 dilution), Notch1 (1:200 dilution) (Novus Biologicals, Littleton, CO, USA); K15 (1:200 dilution) (Thermo Scientific, Waltham, MA, USA); involucrin (1:100 dilution) (Sigma-Aldrich, St. Louis, MO, USA); Ki67 (1:100 dilution) (Dako, Glostrup, CA, USA). Cells were labeled with anti-mouse or anti-rabbit 488 or 546 Alexa Flour secondary antibodies (Invitrogen, Carlsbad, CA, USA). Nuclei were stained with daimidino-2-phenylindole (DAPI; 1:500, Sigma-Aldich, Saint Louis, MO, USA). Samples were analyzed and images recorded using a confocal scanning laser microscope (Leica TCS4D, Leica, Heerbrugg, Switzerland). Quantification of immunofluorescence staining was performed by analyzing six representative fields for each sample by counting the number of positive cells by ImageJ software nCounter plugin (Wayne Rasband, National Institute of Mental Health, Bethesda, MD, USA).

### 4.6. Western Blotting

Total proteins were extracted from KSC and ETA cells freshly isolated by three young children (3–6 years old) or three infants (0–2 years old) foreskin. The samples were analyzed both individually and mixed. Membranes were first incubated overnight at 4 °C with primary antibodies: Beta1 integrin (1:1000, Abcam, Cambridge, UK), ΔNp63 (1:500 dilution, BioLegend, San Diego, CA, USA), Survivin (1:1000, Cell Signaling Technology, Danvers, MA, USA), Involucrin (1:1000, Sigma-Aldrich, St. Louis, MI, USA), K10 (1:1000, Abcam, Cambridge, UK), Claudin 1 (1:1000, Abcam, Cambridge, UK) and β-actin (1:1000, Sigma-Aldrich, St. Loius, MI, USA); than with secondary anti-mouse or anti-rabbit peroxidase-conjugated antibody (1:3000; Bio-Rad Laboratories Inc., Hercules, CA, USA). Protein quantification and western blotting were performed as previously indicated [11,12].

### 4.7. MTT Assay

Freshly isolated keratinocyte subpopulations were seeded in a 96-well tissue culture plate from infants and young children skin and MTT (3-(4,5-dimethylthiazol-2-yl)-2,5-diphenyltetrazolium bromide, Sigma-Aldrich, St. Loius, MI, USA) assay was performed from 24 to 120 h after seeding, as reported [11].

### 4.8. Colony Forming Assay (CFE)

Freshly isolated keratinocyte subpopulations were plated at a density of 100 cells/dish on mitomycin C-treated 3T3 cells and cultivated in DMEM and Ham’s F12 media. Fourteen days later, cells were fixed with 4% formaldehyde and stained with 1% Rhodamine B, as previously reported [11]. Colonies that contained more than 10 cells were counted and CFE was calculated. The colony number was expressed as a percentage of the number of basal cells plated in each dish.

### 4.9. Immunofluorescence and Immunohistochemistry of Skin Equivalent

Skin equivalents were fixed in 4% buffered PFA. Immunohistochemistry of the paraffin-embedded skin equivalent sections was performed according to the UltraVision LP Detection System AP Polymer & Fast Red Chromogen assay (Thermo Fisher Scientific, Waltham, MA, USA). For Immunofluorescence, sections were blocked with a 0.5% BSA/5% goat serum solution for 15 min and then incubated for 1 h at 37 °C with primary antibodies. The corresponding anti-mouse or anti-rabbit 488 or 546 Alexa Flour secondary antibodies (Invitrogen, Carlsbad, CA, USA) were added in the dark for 45 min at room temperature and were analyzed by confocal microscopy (Leica TCS SP2; Leica, Heerbrugg, Switzerland). Primary antibodies utilized were reported in Section 4.5.

The expression intensity was quantitatively determined using ImageJ2 software, version 2.14.0/1.54f (Wayne Rasband, National Institute of Mental Health, Bethesda, MD, USA). This was performed on at least 3 different sections and results were reported. The images are representative of each independent experiment.

## Figures and Tables

**Figure 1 ijms-25-05635-f001:**
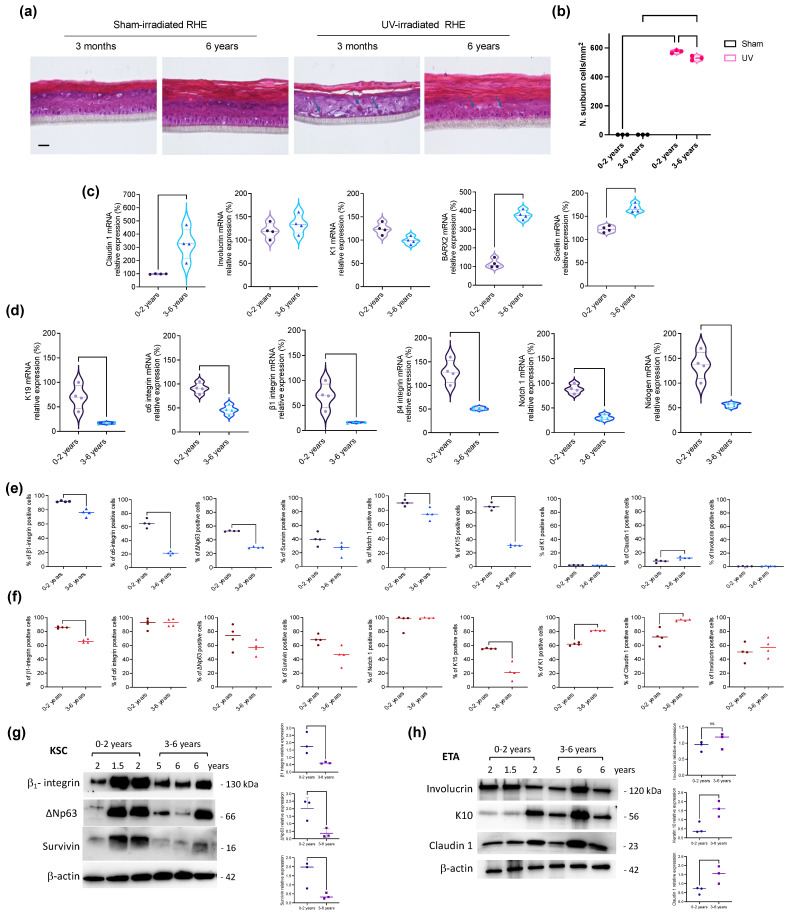
Epidermis in pediatric donors: differentiation of keratinocyte stem cells and their progeny. (**a**) Pictures of representative case of RHE from infants (0–2 years old) and young children (3–6 years old) after sham or UV irradiation. Sunburn cells (arrows) were observed in the UV-irradiated RHE 24 h later. Scale bar 200 µm. (**b**) Sunburn cells/mm^2^ were counted in sham and irradiated RHE (n. 3 samples for each age group). Donor mean age: group 0–2 years old: 0.69 years; group 3–6 years old: 4.67 years. Multiparametric 2-way ANOVA with multiple comparison was used for statistical analysis. (**c**,**d**) RNA was extracted from keratinocyte monolayer cultures derived from n. 4 samples for each age group, and mRNA analysis was performed by qRT-PCR. Donor mean age: group 0–2 years old: 0.73 years; group 3–6 years old: 4.25 years. (**c**) mRNA relative expression of epidermal differentiation and skin barrier genes (Claudin 1, Involucrin, Keratin 1, BARX2 and Sciellin) and (**d**) mRNA relative expression of the interfollicular epidermal stem cell genes (Keratin 19, α_6_-integrin, β_1_-integrin, β_4_-integrin, NOTCH 1 and Nidogen). (**e**) KSC and (**f**) ETA cells from n. 4 young children foreskin or n. 4 infant foreskin were fixed immediately after isolation and spotted onto glass slides, stained with primary antibody and labeled with respectively anti-mouse 488 or anti-rabbit 546 Alexa Flour secondary antibodies. Nuclei were stained with DAPI. The quantification of IF staining was performed by counting positive cells on six representative fields for each sample. Scores were made by means of cell counting ± SD. Donor mean age: group 0–2 years old: 0.73 years; group 3–6 years old: 3.75 years. Mann–Whitney *t*-test was used for statistical analysis. Proteins were extracted from (**g**) KSC and (**h**) ETA cells were freshly isolated from infants (0–2 years old) or young children (3–6 years old) foreskin and lysed for Western blot analysis. Ages are indicated. Donor mean age: group 0–2 years old: 1.83 years; group 3–6 years old: 5.67 years. Images and relative protein quantity of immunoblotting (3 samples/group) show the expression of stemness (β1-integrin, ΔNp63 and Survivin) and differentiation-related markers (Involucrin, K10 and Claudin-1). β-actin was used as normalizing protein by ImageJ software version 2.14.0/1.54f. Student’s *t*-test was used for statistical analysis. ns = Non-Significant.

**Figure 2 ijms-25-05635-f002:**
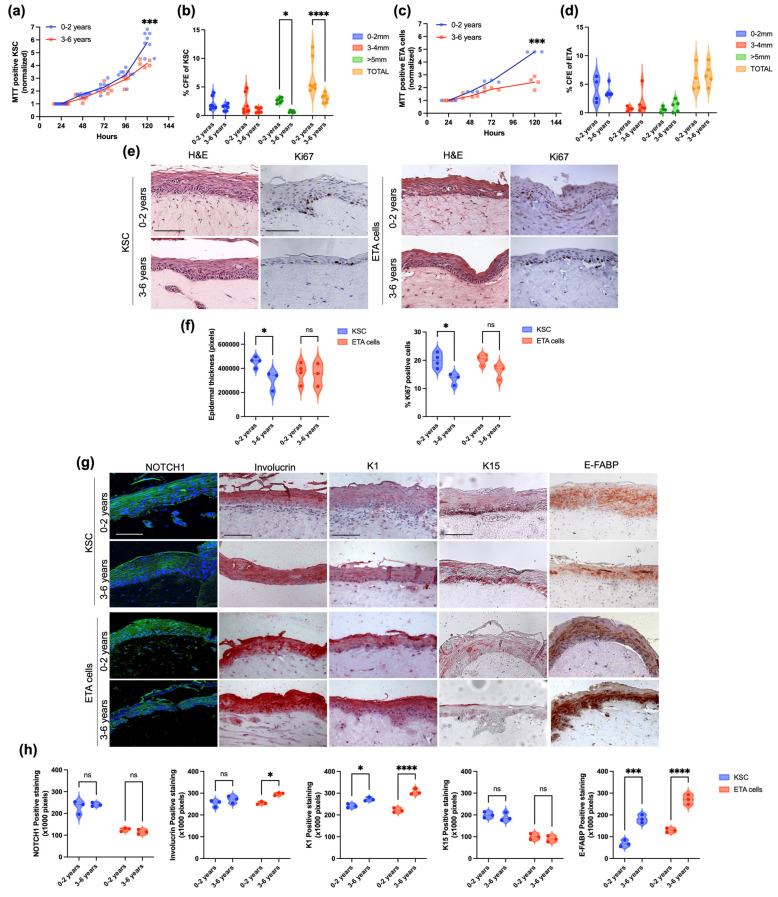
Growth and tissue-regenerative capacity of KSC and ETA cells obtained from both infant and young children skin. (**a**) KSC from n. 8 infants and n. 5 young children foreskin and (**c**) ETA cells from n. 3 infants and n. 4 young children foreskin were seeded in a 96-multiwell plate (5000 cells/well) and MTT assay was performed until 120 h of culture. Donor mean age: group 0–2 years old: 1.69 years; group 3–6 years old: 4.00 years. (**b**) Freshly isolated KSC from n. 7 infants and n. 6 young children foreskin and (**d**) ETA cells from n. 4 infants and n. 5 child foreskin were plated at a density of 100 cells/dish on mitomycin C-treated 3T3 (2.4 × 10^4^/cm^2^) to evaluate their colony forming capacity (CFE). After 14 days, cells were fixed with 4% formaldehyde and stained with 1% Rhodamine B. The colony number was expressed as a percentage of the number of basal cells plated in each dish. Donor mean age: group 0–2 years old: 1.69 years; group 3–6 years old: 4.33 years. Multiparametric 2-way ANOVA and multiple comparison was used for statistical analysis (* 0.01 < *p* < 0.05) (**e**) Skin reconstructs from KSC and ETA cells were generated by seeding freshly isolated keratinocyte subpopulation on dermal equivalent derived from n. 4 infants and n. 3 young children foreskin. Donor mean age: group 0–2 years old: 1.75 years; group 3–6 years old: 4.67 years. After 12 days of sub-merged condition, skin reconstructs were fixed with formalin and sections (4 μm) were stained with H&E and Ki67. Scale bar 200 μm. (**f**) Epidermal thickness was measured by ImageJ software. The number of Ki67 positive cells was measured by ImageJ software nCounter plugin and expressed in percentage of positive cells. Multiparametric 2-way ANOVA and multiple comparison was used for statistical analysis (* 0.01 < *p* < 0.05) (**g**) The expression of NOTCH1, K15, Involucrin, K1 and E-FABP were evaluated in KSC and ETA cell skin reconstructs by IF and IHC (n = 3 samples for age group). The images are representative of each independent experiment. Scale bar 200 μm. (**h**) The expression intensity was quantitatively determined using ImageJ software. This was performed on at least 3 different sections and results were reported. Multiparametric 2-way ANOVA and multiple comparison was used for statistical analysis (* 0.01 < *p* < 0.05; *** *p* < 0.001; **** *p* < 0.0001; ns = Non-Significant).

## Data Availability

Data supporting the findings of this study are available from the corresponding author on reasonable request.

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
