# Peer review of "Characterization of Skin Interfollicular Stem Cells and Early Transit Amplifying Cells during the Transition from Infants to Young Children"

_ijms, 2024, doi:10.3390/ijms25115635_

Round 1

Reviewer 1 Report (New Reviewer)

Comments and Suggestions for Authors

Quadri M et al compared  skin interfollicular stem cells and early  transit amplifying cells at two ages during childhood. The authors found partly obvious differences concerning differentiation and barrier genes, proliferations rates, regenerative capacities and several other skin features.

Major point

1.      

The children were divided into infant (from 1 month to 2 years) and child (from 3 to 6 years). In my opinion, this is arbitrary. Why should there be dramatic differences in skin between an age of two and three years? 

How are “infants” defined?

In sum, it is a child  ≤ 1 year (US, WHO, South Africa and European countries).

·         WHO: A newborn infant, or neonate, is a child under 28 days of age,  Infants (0-1 year of age)

·         www.health.gov.za: Neonates < 28 d; Infant ≥ 28 d - ≤ 1 y; Paediatrics > 1 y - ≤ 14 y; Juvenile > 14 y - ≤ 16 y; Adolescence/Puberty ≤ 19 y

·         U.S. Department of Health and human services/ Centers for Disease control and prevention www.cdc.gov: Infants (0-1 year of age)

Thus, the classification used by the authors is not right and should be corrected. How do the data look like with the correct classification of the skin donors?

It is likely that the investigated skin properties are an age-dependent continuum instead of two different classes. If possible, correlations to age instead of comparison of two arbitrarily chosen donor groups should be shown.

The number of different donors and their age in each figure should always be clear. For all skin/cell samples in each figure the age of the donor should be indicated  (for example all Western blots such as Fig. 1g, h). Furthermore, all data points should be shown (for example fig. 1g, h right panels).

2.      

Concerning this major point: a screening method such as (good old) microarrays or scRNA sequencing data to get transcriptomic profiles of all donor samples and subsequent intelligent bioinformatic analysis will give a good entry point for such an interesting scientific question. Age of the donors can be easily related to transcriptomic profiles and properties of the investigated  cells. This helps to define donor groups, if present, by statistical methods.   

Afterwards, all the methods used and data obtained by the authors are welcome.  

Author Response

Reviewer 2 Report (New Reviewer)

Comments and Suggestions for Authors

Overall assessment:

The strengths of this study are that the quality of the data in the current paper and of the data in reference 10 is generally very good and is based on the work of Jones and Watt Cell 1993. The data are interesting and of value to investigators in the keratinocyte field. The differences between infant and child (juvenile) keratinocyte subpopulations is interesting, novel and relevant to skin biology. There are no major issues with the data quality or its presentation.

The weakness is that the authors equate in vitro stem and transit cell markers with in vivo stem cells which is not really justified in its present form.  The stem cell field has now moved on since the Jones and Watt papers and the clonogenic cells seen in vitro are probably not related to in situ stem cells. Cultured keratinocytes probably represent regenerating cells and so some more introduction and discussion of the work is merited.

There is also evidence that serum free media actually grow suprabasal cells not basal cells (Wilke et al JNCI 1988) although this has not been re-assessed with Ksfm as far as I am aware. Therefore are the authors investigating regenerating or wounded stem cell systems?

This reviewer has not followed the stem cell field for a long time but Jones et al Cell Stem Cell 2007 strongly suggested that there was no such compartment as a transit amplifying cell in the interfollicular epidermis (IFE)and Clevers suggested in a Science review (Clevers Science 2015) there was no such thing as a professional 'stem cell'.

New markers of IFE have recently been identified (Kretzschmar et al Stem Cell Reports 2021) as well as LGR6+ keratinocytes (Füllgrabe et al Stem Cell Reports 2015) and these would be worth introducing and/or discussing as would the recent 'Hallmarks of stemness in mammalian tissues' Beumer and Clevers Cell Stem Cell 2024'. Discussing/introducing these issues would give the article a more up-to-date feel.

Some introductory remarks about BARX2 and Sciellin would also help the non specialised reader as I was not sure how these markers relate to stem and transit amplifying cells.

Detailed Comments:

The term 'child' is not conventional and maybe the authors should consider substituting juvenile for child as I think this is more scientific. Child is a bit vague and I think more of a legal term. 

Results

The experiments do not include positive and negative controls as far as I can see and some of the original blot images do not include the full length of the blots. Could the authors correct this if possible?

Some of the IHC and western data is backed up by qPCR which increases confidence in the data but could the authors make it clear how specific the antibodies are and where this is validated. Commercial antibodies are sometimes not as specific as claimed e.g. the original K15 monoclonal (Aldehlawi et al Sci. Rep. 2019) and the authors have now used a different one. While repeating the whole study would be unreasonable some evidence of antibody specificity should be stated where no q PCR data is provided.

Materials and methods

Line 242 What about the non-adherent cells? what do they do?

Line 246 How long were the keratinocytes cultured in K-sfm? Telomerase which protects basal keratinocytes against ageing switches off within a few days of culture, especially in ksfm and so what the authors' may be analyzing are suprabasal cells in a regenerative state (see also above Wilke et al).

Lines 308-313.  How were the subpopulations released from the collagen? The cloning efficiency looks very low (1-5%) compared to Jones and Watt (Cell 1993) which were essentially 100% when a more gentle procedure was used.

Discussion

Please discuss the findings in the light of the latest stem cell literature.

In particular, it would be good to state why markers such as Lgr6 and LRIG-1 were not studied and relate the authors' findings to these markers and the new stem cell literature.

Comments on the Quality of English Language

Minor Issues

Although the standard of English is excellent, there are some minor grammatical errors such as tense errors and so this should be checked by someone whose first language is English.

Round 2

Reviewer 1 Report (New Reviewer)

Comments and Suggestions for Authors

 Thank you for the revision 1 (R1) of Quadri M et al.: Characterization of skin interfollicular stem cells and early  transit amplifying cells during the transition from infant to child.

 I’m lacking care in the revision to answer and correct the points mentioned by the reviewer.

 Major points

1.   WB figures

Figures 1 g, h) WB: notice in heading of the WB figure the age (in month or years), not the group of the donor used for the original WB. The figure legend is confusing: g) number 3 of young children (3-6 y)   and number 3 of infants (0-2y). What is number 3 (?), it is not indicated in the donor table S1.

 Furthermore,

-         WB fig. 1 g and h (and fig. S2) are slightly manipulated (the signals were made stronger), use the original figures as in the pdf file “ijms-2957231-original-images (1)” 

-         Show all WBs of the other donors used to make graphics fig. 1g and 1h in the pdf file “ijms-2957231-original-images (1)”. The data points of the various donors in one group are uniform (as normally WBs are not).  

-         WB figure S2. Please indicate in the heading of the original WB the age of the donors, not only the group. Indicate in all graphics each data point as you did for figures 1g and 1h. As you can see in your WB original file, the donors are very different (this is ok, but why are the data points so uniform in fig. 1g and fig. 1h?)

2.   Donor groups

The authors added table S1 concerning the skin donors enrolled in this study. This table could be greatly improved: 

-         Why did SK005 appear twice in 0-2 years?

-         The authors should arrange the donors with increasing age in each group. Furthermore, they should add columns for each experiment (column skin equivalent, column WB, column RT-PCR…) and indicate mean age for each experiment in both groups. Then it is clearer that very different groups had been partly compared.

-         For example: The mean age for RT-PCR is in the group (0-2y) 6 month (=0.5 years) and in the group (3-6y) 45 month (=3.75 years). For MTT the mean age is in the group (0-2y) 21 month (=1.75 years) and in the group (3-6y) 48 month (=4 years).

-         If the authors were to compare donor group (0-2y) with mean age 2.1 years with the group (3-6y) with a mean age 3.75 years in qRT-PCR, it is very likely that these groups will not be different.

 In sum, indicate where possible the (mean) age of the donors. Show in the graphics each data point. Add in the pdf file “ijms-2957231-original-images (1)” all WBs used for any figure.

 3.     Add a paragraph limitation of the study in which you address the point of the age of the donors.

Round 3

Reviewer 1 Report (New Reviewer)

Comments and Suggestions for Authors

Thank you for the revision 1 (R1) of Quadri M et al.: Characterization of skin interfollicular stem cells and early  transit amplifying cells during the transition from infant to child.

I’m lacking care in the revision to answer and correct the points mentioned by the reviewer.

Major points

1.   WB figures

Figures 1 g, h) WB: notice in heading of the WB figure the age (in month or years), not the group of the donor used for the original WB. The figure legend is confusing: g) number 3 of young children (3-6 y)   and number 3 of infants (0-2y). What is number 3 (?), it is not indicated in the donor table S1.

Furthermore,

-         WB fig. 1 g and h (and fig. S2) are slightly manipulated (the signals were made stronger), use the original figures as in the pdf file “ijms-2957231-original-images (1)” 

-         Show all WBs of the other donors used to make graphics fig. 1g and 1h in the pdf file “ijms-2957231-original-images (1)”. The data points of the various donors in one group are uniform (as normally WBs are not).  

-         WB figure S2. Please indicate in the heading of the original WB the age of the donors, not only the group. Indicate in all graphics each data point as you did for figures 1g and 1h. As you can see in your WB original file, the donors are very different (this is ok, but why are the data points so uniform in fig. 1g and fig. 1h?)

2.   Donor groups

The authors added table S1 concerning the skin donors enrolled in this study. This table could be greatly improved: 

-         Why did SK005 appear twice in 0-2 years?

-         The authors should arrange the donors with increasing age in each group. Furthermore, they should add columns for each experiment (column skin equivalent, column WB, column RT-PCR…) and indicate mean age for each experiment in both groups. Then it is clearer that very different groups had been partly compared.

-         For example: The mean age for RT-PCR is in the group (0-2y) 6 month (=0.5 years) and in the group (3-6y) 45 month (=3.75 years). For MTT the mean age is in the group (0-2y) 21 month (=1.75 years) and in the group (3-6y) 48 month (=4 years).

-         If the authors were to compare donor group (0-2y) with mean age 2.1 years with the group (3-6y) with a mean age 3.75 years in qRT-PCR, it is very likely that these groups will not be different.

In sum, indicate where possible the (mean) age of the donors. Show in the graphics each data point. Add in the pdf file “ijms-2957231-original-images (1)” all WBs used for any figure.

3.     Add a paragraph limitation of the study in which you address the point of the age of the donors.

 Thank you for the revision 1 (R1) of Quadri M et al.: Characterization of skin interfollicular stem cells and early  transit amplifying cells during the transition from infant to child.

I’m lacking care in the revision to answer and correct the points mentioned by the reviewer.

Major points

1.   WB figures

Figures 1 g, h) WB: notice in heading of the WB figure the age (in month or years), not the group of the donor used for the original WB. The figure legend is confusing: g) number 3 of young children (3-6 y)   and number 3 of infants (0-2y). What is number 3 (?), it is not indicated in the donor table S1.

Furthermore,

-         WB fig. 1 g and h (and fig. S2) are slightly manipulated (the signals were made stronger), use the original figures as in the pdf file “ijms-2957231-original-images (1)” 

-         Show all WBs of the other donors used to make graphics fig. 1g and 1h in the pdf file “ijms-2957231-original-images (1)”. The data points of the various donors in one group are uniform (as normally WBs are not).  

-         WB figure S2. Please indicate in the heading of the original WB the age of the donors, not only the group. Indicate in all graphics each data point as you did for figures 1g and 1h. As you can see in your WB original file, the donors are very different (this is ok, but why are the data points so uniform in fig. 1g and fig. 1h?)

2.   Donor groups

The authors added table S1 concerning the skin donors enrolled in this study. This table could be greatly improved: 

-         Why did SK005 appear twice in 0-2 years?

-         The authors should arrange the donors with increasing age in each group. Furthermore, they should add columns for each experiment (column skin equivalent, column WB, column RT-PCR…) and indicate mean age for each experiment in both groups. Then it is clearer that very different groups had been partly compared.

-         For example: The mean age for RT-PCR is in the group (0-2y) 6 month (=0.5 years) and in the group (3-6y) 45 month (=3.75 years). For MTT the mean age is in the group (0-2y) 21 month (=1.75 years) and in the group (3-6y) 48 month (=4 years).

-         If the authors were to compare donor group (0-2y) with mean age 2.1 years with the group (3-6y) with a mean age 3.75 years in qRT-PCR, it is very likely that these groups will not be different.

In sum, indicate where possible the (mean) age of the donors. Show in the graphics each data point. Add in the pdf file “ijms-2957231-original-images (1)” all WBs used for any figure.

3.     Add a paragraph limitation of the study in which you address the point of the age of the donors.

Author Response

We are very thankful to the reviewer for the suggestion, allowing us to improve our manuscript. Please see below a point-by-point answers (in blue) to the reviewer’s comments.

Major points

  1. WB figures

Figures 1 g, h) WB: notice in heading of the WB figure the age (in month or years), not the group of the donor used for the original WB. The figure legend is confusing: g) number 3 of young children (3-6 y)   and number 3 of infants (0-2y). What is number 3 (?), it is not indicated in the donor table S1.

Thank you. We definitely consider this WB not enough clear, given that we analyzed a mixture of 3 samples instead of loading them separately. Thus, we decided to delete and substitute it with the WB already presented in Supplementary material in which we have 3 samples for each group in the same blot. Moreover, we also analyze the relative amount of protein for each marker and show in graphics each data point corresponding to each donor. Moreover, we also added in the WB heading the age of each donor.

 Furthermore,

-         WB fig. 1 g and h (and fig. S2) are slightly manipulated (the signals were made stronger), use the original figures as in the pdf file “ijms-2957231-original-images (1)” 

We amended the WB figure and use the original-not contrasted images. Please consider that the figure presented in supplementary material as original images are composite images obtained merging the chemiluminescent and colorimetric picture of the blot.

-         Show all WBs of the other donors used to make graphics fig. 1g and 1h in the pdf file “ijms-2957231-original-images (1)”. The data points of the various donors in one group are uniform (as normally WBs are not).  

As mentioned above, we amended the figure 1 g) and h) because we decided to delete and substitute it with the WB already presented in Supplementary material in which we have 3 samples for each group in the same blot. Moreover, we also analyze the relative amount of protein for each marker and show in graphics each data point corresponding to each donor. Original images are presented accordingly in the supplementary material.

-         WB figure S2. Please indicate in the heading of the original WB the age of the donors, not only the group. Indicate in all graphics each data point as you did for figures 1g and 1h. As you can see in your WB original file, the donors are very different (this is ok, but why are the data points so uniform in fig. 1g and fig. 1h?)

We amended the figure accordingly, and now the WB formerly present in S2 is in Figure 1g and h, please see above. Moreover, we also indicate in the figure legend the mean age of each group donor.

  1. Donor groups

The authors added table S1 concerning the skin donors enrolled in this study. This table could be greatly improved: 

We improved the table accordingly to reviewer’s comments (now presented as the new Supplementary table 1 in the supplementary material file).

Please see below our

-         Why did SK005 appear twice in 0-2 years?

Thank you for pointing out this matter. We are sorry for this. Indeed, we switched two numbers while filling in the table, now it is corrected.

-         The authors should arrange the donors with increasing age in each group. Furthermore, they should add columns for each experiment (column skin equivalent, column WB, column RT-PCR…) and indicate mean age for each experiment in both groups. Then it is clearer that very different groups had been partly compared. For example: The mean age for RT-PCR is in the group (0-2y) 6 month (=0.5 years) and in the group (3-6y) 45 month (=3.75 years). For MTT the mean age is in the group (0-2y) 21 month (=1.75 years) and in the group (3-6y) 48 month (=4 years).

We thank the reviewer for the greater input. We amended the Table accordingly (we decided to indicate age donor only in years) and now we think it is really clearer and more straightforward.

-         If the authors were to compare donor group (0-2y) with mean age 2.1 years with the group (3-6y) with a mean age 3.75 years in qRT-PCR, it is very likely that these groups will not be different.

We calculated the mean age of each group in every presented assay. Differences among the 2 groups are evident. We amended each figure legend accordingly. i.e. for qRT-PCR the mean age of donor group (0-2 years) is 0.73 years while the mean age of donor group (3-6 years) is 4.25 years, supporting the differences of the 2 groups.

 In sum, indicate where possible the (mean) age of the donors. Show in the graphics each data point. Add in the pdf file “ijms-2957231-original-images (1)” all WBs used for any figure.

We amended figure legends accordingly.

  1. Add a paragraph limitation of the study in which you address the point of the age of the donors.

We inserted a brief extract on this topic in the discussion paragraph, please see page 7, line 353-357.

This manuscript is a resubmission of an earlier submission. The following is a list of the peer review reports and author responses from that submission.

Round 1

Reviewer 1 Report

Comments and Suggestions for Authors

In this manuscript, Quadri et al. characterize keratinocytes stem cells (KSC) and early transit amplifying (ETA) cells from interfollicular epidermis during the transition from infant to child skin. The authors indicate that KSC from infants possess more regenerative capacity than KSC from children. In addition, the authors also indicate that ETA cells from infants express lower levels of differentiation markers compared with children-ETA cells, which correlate with immature skin barrier functions in infant skin. The presented work is potentially important for dermatology, relevant to the scope of Int. J. Mol. Sci., and will be of interest to its readership. However, there are some issues that need to be addressed as detailed below.

(1) Figure 1b: There is no indication of the number of separate experiments performed (i.e., the “n”) nor quantitation of the results. The authors should provide the n, quantify the sections and show cumulative results, along with statistical analysis of the results.

(2) Figure 2g and S1: Staining patterns for K1, K15 (fig 2g), and ΔNp63 (fig S1) seem to be strange. K1 expression should be widely observed in suprabasal epidermal keratinocytes, whereas K15 expression should be observed in basal layer of the epidermis (e.g., Figure 1 in PMID: 30456360; they also use RHE for immunostaining against K1 and K15). However, RHE originated from KSC does not express K1 at all. In addition, K15 staining is not restricted to the basal layer of the RHE. Furthermore, though the authors’ data show that ΔNp63 staining is mostly observed in cytoplasm, ΔNp63 staining should be observed in nucleus.

(3) Although the authors use Notch1 as the stemness/proliferation markers, Notch1 in epidermis functions as inducer of differentiation, rather than maintenance of stemness (e.g. PMID: 11432830, 17079689, 18410734).

Author Response

In this manuscript, Quadri et al. characterize keratinocytes stem cells (KSC) and early transit amplifying (ETA) cells from interfollicular epidermis during the transition from infant to child skin. The authors indicate that KSC from infants possess more regenerative capacity than KSC from children. In addition, the authors also indicate that ETA cells from infants express lower levels of differentiation markers compared with children-ETA cells, which correlate with immature skin barrier functions in infant skin. The presented work is potentially important for dermatology, relevant to the scope of Int. J. Mol. Sci., and will be of interest to its readership. However, there are some issues that need to be addressed as detailed below.

(1) Figure 1b: There is no indication of the number of separate experiments performed (i.e., the “n”) nor quantitation of the results. The authors should provide the n, quantify the sections and show cumulative results, along with statistical analysis of the results.

        We agree with the reviewer that there are no indications of the number of separate experiments performed in Figure 1b. We clarify that the images in figure 1a are representative of each group of age (1-2 years and 3-6 years), we performed the experiment in triplicate, quantified every single section, and reported the cumulative results in figure 1b. In the revised version of the manuscript, we indicated the number of samples in the figure legend as follows: 1-2 years (3 samples), 3-6 years (3 samples) (please, see figure 1b).

(2) Figure 2g and S1: Staining patterns for K1, K15 (fig 2g), and ΔNp63 (fig S1) seem to be strange. K1 expression should be widely observed in suprabasal epidermal keratinocytes, whereas K15 expression should be observed in basal layer of the epidermis (e.g., Figure 1 in PMID: 30456360; they also use RHE for immunostaining against K1 and K15). However, RHE originated from KSC does not express K1 at all. In addition, K15 staining is not restricted to the basal layer of the RHE. Furthermore, though the authors’ data show that ΔNp63 staining is mostly observed in cytoplasm, ΔNp63 staining should be observed in nucleus.

    We thank the reviewer for allowing us to improve our manuscript. We agreed that the images of skin reconstructs reporting the expression of K15 and K1 are not representative and we have carefully replaced them with those more representatives. However, we would like to clarify that the localization of keratins in RHE differs from human skin in vivo. In fact, skin reconstructs produced in vitro are governed by a bigger keratinocyte proliferative boost. In healthy skin, K15 is expressed in the basal layer by keratinocyte stem cells (KSC). In our model, K15 is expressed in the basal layer of RHE derived from KSC cells, but not specifically observed in the basal layer of skin equivalent derived from ETA cells. This different localization is due to the more differentiation state of RHE derived from ETA cells. In fact, as reported by Li et al (PMID: 14755336), the organotypic cultures derived from the ETA fraction exhibit sporadic re-expression of K15 in the basal layer. Furthermore, our results show that K15 is barely detectable in child ETA-derived RHE, suggesting that ETA derived from child skin are more differentiated compared to those derived from infant skin (Please, see new figure 2g, and Lotti et al., 2022 PMID: 36037263).

On the other hand, K1 is generally expressed during the first phases of the differentiation process. In our work, K1 is barely expressed by RHE derived from KSC cells due to their lower differentiation state.

Moreover, we cannot compare an RHE produced by starting from a suspension of bulk (total) keratinocytes, having different differentiation state conditions, with an RHE derived from KSC or ETA, which are performed starting from a specific keratinocyte subpopulation. For this reason, we cannot compare our RHEs with those described in PMID: 30456360.

As concern for the localization of ΔNp63, Albasri and co-workers showed that the cytoplasmic expression of p63 seems like to be related to a high expression of Ki67 in colorectal cancer (PMID: 3105618), suggesting that the cytoplasmatic expression of ΔNp63 could be correlated to a high proliferation rate in tumors. Furthermore, to confirm the levels of ΔNp63 in relation to KSC proliferation, we evaluated its expression in KSC derived from infant (1-2 years) and child (3-6 years) skin sample by western blotting (please see figure 1g).

(3) Although the authors use Notch1 as the stemness/proliferation markers, Notch1 in epidermis functions as inducer of differentiation, rather than maintenance of stemness (e.g. PMID: 11432830, 17079689, 18410734).

   The are several conflicting data about the role of Notch1 in the epidermis. Recently, we showed that Notch1 maintains stemness in human keratinocytes via bi-directional crosstalk with surviving, suggesting that it could be used as a stemness marker in keratinocytes (Please, see Palazzo et al., 2015 PMID: 26540052)

Reviewer 2 Report

Comments and Suggestions for Authors

The authors provide a brief characterisation of interfollicular keratinocyte stem cells and early transit amplifying cells from infant skin as compared to child skin. They used studies of proliferation and gene expression in cultured cells and reconstructed epidermis/skin. This is of some interest as it highlights changes in epidermal function occurring during the first years after birth. The findings are mostly descriptive and the main concern is that the conclusions in the discussion and elsewhere are too speculative. Two different stages were tested, keratinocytes from infant skin <2y and child skin at 3-6y. Conclusions about changes "with age" should be made carefully.

More importantly, conclusions about "the correlation between age and structural maturation of the skin barrier" are not justified; neither the structure of the skin barrier nor skin barrier function were analysed here. The conclusion is mainly based on expression of 5 genes involved in keratinocyte differentiation, or even 2 genes in case of the analysis of reconstructed skin. This should be taken into account for discussing the results; therefore a sentence such as "...indicating that the skin barrier develops and organizes with age" does not seem to be appropriate.

Other comments:

1. Abstract: "differentiation and barrier [?] genes increase with age" should probably refer to gene expression.

2. "...proliferation rate...is higher in infants than in children..." and similar instances: should refer to infant and children skin samples.

3. "...proliferation rate...is higher in infants than in children, while the capacity...": the contrast is not clear here, both findings are more prominent in infant samples.

4. Results: "mRNA expression of genes..." should be just "expression of genes".

5. "involucrin and claudin 1 were expressed only in ETA cells": Why is this expression so strong?

6. "in particular, involucrin and K1 expression was significantly more pronounced...": Only involucrin and K1 are shown, were other differentiation markers tested as well?

6. Methods: Only GAPDH was used as a reference gene; is there any evidence that this is suitable for these cells that seem to undergo significant developmental changes?

7. Please indicate all numbers of replicates used in these experiments.

Author Response

The authors provide a brief characterisation of interfollicular keratinocyte stem cells and early transit amplifying cells from infant skin as compared to child skin. They used studies of proliferation and gene expression in cultured cells and reconstructed epidermis/skin. This is of some interest as it highlights changes in epidermal function occurring during the first years after birth. The findings are mostly descriptive and the main concern is that the conclusions in the discussion and elsewhere are too speculative. Two different stages were tested, keratinocytes from infant skin <2y and child skin at 3-6y. Conclusions about changes "with age" should be made carefully.

More importantly, conclusions about "the correlation between age and structural maturation of the skin barrier" are not justified; neither the structure of the skin barrier nor skin barrier function were analysed here. The conclusion is mainly based on expression of 5 genes involved in keratinocyte differentiation, or even 2 genes in case of the analysis of reconstructed skin. This should be taken into account for discussing the results; therefore a sentence such as "...indicating that the skin barrier develops and organizes with age" does not seem to be appropriate.

We are very thankful to the reviewer for giving us the possibility to improve our manuscript. Our study was focused on the evaluation of the proliferative and regenerative ability of keratinocytes progenitors during infancy, and the analysis of those factors strictly related to stemness, differentiation and skin barrier functionality.  In this context, the use of skin reconstructs allow to understand the complexity of these processes in a timely-dependent manner. In fact, its formation includes a step-by-step assembling of progressively differentiating keratinocytes, which terms with the creation of a complete, air-exposed, skin (Please see PMID: 25939812; PMID: 19908015; PMID: 25330297; PMID: 36037263).  Therefore, by using skin equivalents, we carefully defined the development of epidermis starting from keratinocytes progenitor (KSC and ETA) cells of different age (from 0 to 6 years old) and with different differentiative state. Thanks to this model, we obtained a tool to evaluate the development and formation of skin barrier in correlation to skin homeostasis. Moreover, we can also justify the correlation between age and structural maturation of the skin barrier by the analysis of the resistance to UV-induced damage (Figure 1a). In addition, we added the evaluation of the Epidermal Fatty Acid Binding Protein (E-FABP) in skin reconstructs, which is a well-known marker of skin barrier and terminal differentiation (Please see PMID: 11874481; PMID: 35445019; PMID: 23528210). Finally, we considered the suggestion of the reviewer, and reworded the discussion accordingly.

Other comments:

  1. Abstract: "differentiation and barrier [?] genes increase with age" should probably refer to gene expression.

Yes, we referred to gene expression (mRNA) of genes related to differentiation, skin barrier and stemness. We apologize for this error, and we changed the sentence accordingly (please see the abstract and results 2.1)

  1. “…proliferation rate…is higher in infants than in children…” and similar instances: should refer to infant and children skin samples.

We referred to cells derived from infant and children skin samples. We changed the sentence to make it clearer to the reader (please see the abstract and result 2.2).

  1. "...proliferation rate...is higher in infants than in children, while the capacity...": the contrast is not clear here, both findings are more prominent in infant samples.

We agree with the reviewer that there is no contrast in the sentence. We changed it accordingly (please see the abstract).

  1. Results: "mRNA expression of genes..." should be just "expression of genes".

We agree with the reviewer that this sentence is wrong, and we changed it as suggested (please see the results section, line 67).

  1. "involucrin and claudin 1 were expressed only in ETA cells": Why is this expression so strong?

We are very thankful to the reviewer because we realized that the images for Involucrin staining were not representative. So, we carefully revised them and decided to change them with those more representatives (please see supplementary materials).

The “right” expression of Claudin-1 ETA cells, in terms of intensity, is unknown. Claudin-1 is a protein of the tight junction, which starts to be expressed in the first phases of keratinocyte differentiation, of which ETA cells are an essential part as a KSC progenitor. On the contrary, given that involucrin is a marker of late differentiation, it is less expressed in ETA cells than Claudin-1.

  1. "in particular, involucrin and K1 expression was significantly more pronounced...": Only involucrin and K1 are shown, were other differentiation markers tested as well?

We evaluated the expression of K1 and Involucrin because they are commonly utilized to describe early and late differentiation (PMID: 10688377, PMID: 21957444). Also, we evaluated the expression of Claudin-1 in KSC and ETA cells and the expression of several genes correlated to epidermal differentiation (i.e. Claudin-1, BARX2 and Sciellin). Moreover, we added the evaluation of the Epidermal Fatty Acid Binding Protein (E-FABP) in skin reconstruct and the evaluation of Keratin 10 and involucrin by western blotting (please see figure 2g-h and figure 1h, respectively).

  1. Methods: Only GAPDH was used as a reference gene; is there any evidence that this is suitable for these cells that seem to undergo significant developmental changes?

GAPDH gene expression seems to change in response to mitogenic stimuli or after UV-B treatment (PMID: 19209154). In this work, cells were cultured and lysed without mitogenic stimuli. Anyway, we also used the ribosomal RNA S28 (RPS289) as a reference gene, and no differences were observed in gene expression. We added more clarification in the materials and methods section.

  1. Please indicate all numbers of replicates used in these experiments.

We agree with the reviewer that we did not indicate the number of replicates, and we apologize for that. Now, we added the replicates in each figure or figure legend.

Round 2

Reviewer 1 Report

Comments and Suggestions for Authors

Improved.

Author Response

-

Reviewer 2 Report

Comments and Suggestions for Authors

Authors have made several amendments and improved the manuscript; most comments were taken into account. I still don't agree with the conclusions about barrier function. I agree with the response regarding reconstructed skin, it is suitable to analyse barrier function but this has not been done here. It is appreciated that analysis of another differentiation marker, E-FABP, has been added.

However, this does not add to the understanding of barrier function; while details of E-FABP function are not well understood, the publications referred to in the answer letter do not describe the role in barrier function. PMID 11874481 describes a ko mouse model with minor changes to barrier function and no mechanistic insight, PMID 23528210 does not describe the barrier at all but concludes that "E-FABP plays an important role in keratinocyte differentiation". This is quite clear and E-FABP could be referred to as a differentiation marker. Epidermal differentiation is not identical though with barrier function; for instance, ko animals and human diseases demonstrate this. This connection should be reduced, as has been done partly in the Discussion. Similarly, in a conclusion such as "The study presents a novel insight into pediatric skin, and sheds light on the correlation between age and structural maturation of the skin barrier", barrier should be removed.

Author Response

We agree with the reviewer that, in our work, a study of skin barrier functionality, such as permeability and water loss, is missing. However, it is known that E-FABP is responsible for the water permeability barrier of the skin (PMID: 12479572). Nevertheless, as suggested by the reviewer, we decided to remove the barrier function from our conclusion (please see the abstract).